# A deep learning-based workflow for high-throughput and high-quality widefield fluorescent imaging of 3D samples

**Edvin Forsgren**[*1]                                          edvin.forsgren@umu.se

**Christoffer Edlund**[*2]                          Christoffer.Edlund@Sartorius.com

**Miniver Oliver**[3]                                  Minnie.Oliver@Sartorius.com

**Kalpana Barnes**[3]                              Kalpana.Barnes@Sartorius.com

**Timothy R Jackson**[3]                          Timothy.Jackson@Sartorius.com

**Rickard Sjögren**[1,2]                          Rickard.Sjoegren@Sartorius.com

[1] *Computational Life Science Cluster, Umeå University, Umeå, Sweden*

[2] *Sartorius Corporate Research, Umeå, Sweden*

[3] *Sartorius, BioAnalytics, Royston, United Kingdom*

## Abstract

3D widefield fluorescent microscopic (wFLM) imaging is a widespread technology used to study living three-dimensional samples, such as tumor spheroids during drug development. However, 3D wFLM imaging suffers from a severe trade-off between image quality and throughput limiting its applicability. In this project, we present a novel workflow that enables high-throughput 3D wFLM imaging, which has previously been impossible, and apply it to fluorescent indicators of cell health within 3D tumor spheroids. The workflow combines deep learning with state-of-the-art live-cell imaging techniques to speed up the acquisition of a fluorescent image of a three-dimensional sample by a factor of a hundred.

**Keywords:** Convolutional Neural Network, Fluorescent 3D cell imaging, Conditional Generative Adversarial Network, Fluorescent live-cell imaging, High-throughput, Z-sweep, OSA-blocks, Tumor Spheroids

## 1. Introduction

3D widefield fluorescent imaging of tumor spheroids is challenging because in order to acquiring images of the entire sample, many images must be acquired at different focal depths to obtain a so-called Z-stack. Z-stack acquisitions have two major drawbacks: visualizing large samples can be very time-consuming, and the longer cells are exposed to fluorescent light the greater the risk of phototoxicity and photobleaching. These problems become more severe when samples are analyzed over time, requiring repeated Z-stack acquisitions, limiting the overall throughput capabilities of live-cell 3D widefield fluorescent imaging. An alternative to the Z-stack would be an image acquired by opening the shutter and moving the camera along the Z-axis through the entire 3D sample, producing a blurry image of fluorescence integrated over the Z-dimension. This image, which we are calling a Z-sweep, can be as much as a hundred times faster to acquire than the traditional Z-stack method. We aim to combine these imaging techniques with state-of-the art deep learning methods to solve the issue of high-throughput and high-quality widefield fluorescent imaging of 3D samples.

---

[*] Contributed equally

## 2. Methods

We present a workflow combining fast Z-sweep acquisition and convolutional neural network (CNN)-based image processing. We then train a Convolutional Neural Network (CNN) to predict a high-quality 2D-projection calculated from a Z-stack based on the corresponding blurry Z-sweep image by minimizing the L1-difference norm between the predicted projection compared to the true one. We test four different CNNs based on the U-net architecture (Ronneberger et al., 2015) (Figure 1). The CNN that performed the best is a modified variant of U-net that we call OSA-U-net, which uses one-shot aggregation (OSA) and effective Squeeze-and-Excitation blocks (Lee and Park, 2020). Additionally, we trained the CNNs using a second discriminator CNN in a conditional adversarial generative adversarial network (cGAN)-fashion (Isola et al., 2017) to improve the predicted projection appearance.

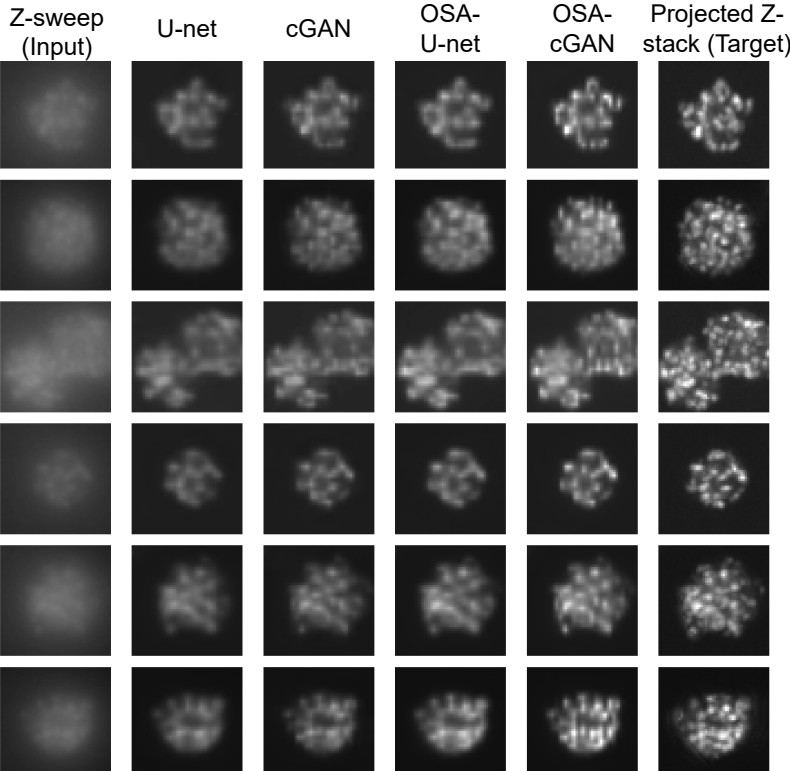

Figure 1: Example of cell clusters from the multi-spheroid images predictions from different CNNs.

## 3. Results

We find that our CNN reliably predicts high-quality projection images based on the blurry Z-sweeps (Figure 1 and Table 1), providing a way for both high-throughput and high-quality 3D fluorescent imaging. Our CNN, OSA-U-Net cGAN with a PatchGAN discriminator (Isola et al., 2017), achieves the best predictions in visual quality as evaluated by the peak

signal to noise ratio and Frechet inception distance (Heusel et al., 2018) for both single- and multi-spheroid assays compared to the baselines. We also validated fluorescent signal intensity trends from the Z-sweep/CNN-based workflow against the traditional projected Z-stack measurements, meaning that we can draw the same biological conclusions as when using the much more time-consuming Z-stack acquisition ($R^2 = 0.99443$). This is further exemplified in an experiment where embedded tumor multi-spheroids are treated with cytotoxic compounds, and we can use the Z-sweep/CNN-based workflow to quantify reduced cell health over time in response to drug treatment.

Table 1: Comparing FID, PSNR and SSIM ($\pm$ one standard deviation) of different networks evaluated on unseen spheroid images. Smaller FID means more perceptually similar, i.e. better.

|  | FID | PSNR | SSIM |
|---|---|---|---|
| projection vs sweep | 294.25 | 17.97 $\pm$ 4.05 dB | 0.075 $\pm$ 0.045 |
| OSA-CGAN | 99.07 | 23.66 $\pm$ 4.8 dB | 0.107 $\pm$ 0.041 |
| CGAN | 121.47 | 20.44 $\pm$ 5.74 dB | 0.089 $\pm$ 0.037 |
| OSA-U-Net | 113.59 | 22.45 $\pm$ 6.39 dB | 0.109 $\pm$ 0.041 |
| U-Net | 105.47 | 21.02 $\pm$ 5.64 dB | 0.081 $\pm$ 0.035 |

## 4. Conclusion

To conclude, we show how fast Z-sweep acquisition combined with state-of-the-art deep learning methodologies provide a way to acquire high-quality fluorescent images of 3D-samples with 10-100 times higher throughput than the standard Z-stack acquisition.

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
