# OpenReview forum: "A deep learning-based workflow for high-throughput and high-quality widefield fluorescent imaging of 3D samples"
_MIDL.io/2021/Conference/Short — Submitted to MIDL 2021_

### Official Review · Reviewer_SfMY · 2021-04-30

**Confidence:** 3
**Final Rating:** 3

**Summary:**

The authors present a workflow to speed up the image acquisition process for 3D widefield fluorescent imaging. Instead of acquiring images at different focal depths, which is time-consuming, they open the camera's shutter and move the camera along the z-axis. From this resulting blurry image, CNNs reconstruct the desired 3D sample.

**Strengths:**

In most parts, the paper is very well written. The idea and the advantages of the method are clearly stated. The authors provide an extensive evaluation with four different CNNs. Additional experiments weaken the significant concern that the resulting images may look nice but cannot be used for biological diagnostics.

**Weaknesses:**

Although one can assess the PSNR and FID results as good enough, the structural image similarity values seem very low. The values of the original sweep and the best method have overlapping standard deviations. This fact is not discussed any further.

The authors should be more specific when describing the used datasets: How many images are used for training and testing? How do the train and test sets differ? Which 2D projection is used from the Z-stack, are different z-positions evaluated? Furthermore, it is not entirely clear whether the Z-sweep is calculated from the Z-stack, i.e., simulated, or acquired by the proposed method.

**Deanonymize Review:**

no

**Detailed Comments:**

The authors should also discuss actual numbers: how long does the baseline Z-stack recording take, how long the Z-sweep acquisition and CNN-reconstruction?

**Justification Of The Rating:**

For a short paper, the proposed work offers a substantial evaluation part and is well-written. However, in some sections, the paper should be more specific. The results are not entirely convincing. The method appears to be novel.

**Paper Type:**

methodological development

**Special Issue:**

no

---

### Official Review · Reviewer_MKue · 2021-04-30

**Confidence:** 5
**Final Rating:** 1

**Summary:**

The authors show an application of Deep Learning (DL) models to denoise or filter out a fluorescence microscopy image. The concrete application performs the processing of a 2D image that results from the integral along the Z dimension of a 3D tumor spheroid. They evaluate four different architectures  (UNet, cGAN, OSA-UNet, OSA-cGAN).

**Strengths:**

The authors propose four different Deep Learning approaches (UNet, cGAN, OSA-UNet, OSA-cGAN) and evaluate them on the same data. The results seem to be quite accurate and the numbers are presented together with standard deviations.

**Weaknesses:**

While the proposed architectures are new for this particular application, I feel that the authors did not look for any previous work or state-of-the-art methods. The introduction lacks references in the field of denoising or image restoration.

The work is claimed for "high-throughput and high-quality widefield fluorescent imaging of 3D samples" while the information that is being recovered is only on the integral of the Z-dimension and not really 3D information. For what would be interesting the information in such 2D projections? I think this should be included in the motivation as it is confusing what is the real purpose of the work.

One of the advantages of this method is accelerating image acquisition. Could the authors provide some numbers on it and on the image processing part?

How trustworthy in terms of biological meaning are the results obtained? This is one of the major risks when using DL in microscopy so it's highly recommended to comment on this.

What is the meaning of the acronyms FID, PSNR, and SSIM?

How were the methods trained? what data di you use?

**Deanonymize Review:**

no

**Justification Of The Rating:**

The presented abstract has several important weaknesses such as the citation of previous work or state-of-the-art, the real gaining and the usefulness of the current method, proper description of the image processing workflow.

**Paper Type:**

validation/application paper

**Special Issue:**

no

---

### Meta-Review · Area_Chair_qAeK · 2021-05-07

**Recommendation:** Reject
**Confidence:** 5

**Metareview:**

While the paper describes an interesting approach, I have to agree with reviewer MKue that even in a short paper there should be at least a brief discussion of related work.

---

### Decision · Program_Chairs · 2021-05-11

Reject